# Biochemical Evolution of a Potent Target of Mosquito Larvicide, 3-Hydroxykynurenine Transaminase

**DOI:** 10.3390/molecules27154929

**Published:** 2022-08-02

**Authors:** Huaqing Chen, Biswajit Bhowmick, Yu Tang, Jesus Lozano-Fernandez, Qian Han

**Affiliations:** 1Laboratory of Tropical Veterinary Medicine and Vector Biology, School of Life Sciences, Hainan University, Haikou 570228, China; huaqing.chen@hainanu.edu.cn (H.C.); biswajit.bhowmick@hotmail.com (B.B.); 13527583652@163.com (Y.T.); 2One Health Institute, Hainan University, Haikou 570228, China; 3Department of Genetics, Microbiology and Statistics, Biodiversity Research Institute (IRBio), University of Barcelona, Avd. Diagonal 643, 08028 Barcelona, Spain; jesus.lozanof@gmail.com

**Keywords:** molecular docking, in silico methods, molecular evolution, 3-hydroxykynurenine transaminase, alanine glyoxylate transaminase

## Abstract

A specific mosquito enzyme, 3-hydroxykynurenine transaminase (HKT), is involved in the processing of toxic metabolic intermediates of the tryptophan metabolic pathway. The HKT enzymatic product, xanthurenic acid, is required for *Plasmodium* spp. development in the mosquito vectors. Therefore, an inhibitor of HKT may not only be a mosquitocide but also a malaria-transmission blocker. In this work, we present a study investigating the evolution of HKT, which is a lineage-specific duplication of an alanine glyoxylate aminotransferases (AGT) in mosquitoes. Synteny analyses, together with the phylogenetic history of the AGT family, suggests that HKT and the mosquito AGTs are paralogous that were formed via a duplication event in their common ancestor. Furthermore, 41 amino acid sites with significant evidence of positive selection were identified, which could be responsible for biochemical and functional evolution and the stability of conformational stabilization. To get a deeper understanding of the evolution of ligands’ capacity and the ligand-binding mechanism of HKT, the sequence and the 3D homology model of the common ancestor of HKT and AGT in mosquitoes, ancestral mosquito AGT (AncMosqAGT), were inferred and built. The homology model along with 3-hydroxykynurenine, kynurenine, and alanine were used in docking experiments to predict the binding capacity and ligand-binding mode of the new substrates related to toxic metabolites detoxification. Our study provides evidence for the dramatic biochemical evolution of the key detoxifying enzyme and provides potential sites that could hinder the detoxification function, which may be used in mosquito larvicide and design.

## 1. Introduction

The tryptophan metabolic pathway is evolutionary conserved amongst distantly related groups, such as bacteria, fungi, plants, or animals, showcasing its relevance in most living organisms. Specifically, 3-hydroxykynurenine (3-HK) is an intermediate metabolite in the tryptophan oxidation pathway. In many organisms, 3-HK is a potential endogenous neurotoxin, being dose-sensitive to its presence, as it can stimulate the production of reactive oxygen species, which can lead to cell death [1,2]. In insects, exogenous administration of 3-HK has drastic effects, leading to irreversible paralysis and death [3,4]. This metabolite induces *Plasmodium berghei* infection in *Anopheles* mosquitoes by damaging the structure of the peritrophic matrix in the midgut. Nevertheless, it has been shown that the gut microbiota (i.e., *Pseudomonas alcaligenes*) in *A. stephensi* helps in improving the resistance towards malaria parasites by synthesizing the enzyme kynureninase that catabolizes 3-HK [5,6]. Although the potential toxicity of 3-HK has been highlighted in many studies, it remains present in various physiological functions as a compound of significance. In insects, 3-HK is also an immediate precursor of ommochromes [7,8]. Therefore, maintaining the dynamic balance of this molecule is crucial to protect the organism from being damaged.

Mosquito-borne viral diseases infect more than 700 million people and kill about 700,000 people every year, continuing to spread worldwide and causing severe public health problems [9]. In mosquitoes, the transamination of 3-HK to xanthurenic acid is the major branch pathway of tryptophan metabolism [10]. They metabolize 3-HK using the enzyme 3-hydroxykynurenine transaminase (HKT), which is a lineage-specific paralogous of the alanine glyoxylate aminotransferases (AGTs). Interestingly, mosquitoes are unique in having evolved two AGTs. AGTs are commonly involved in the glyoxylate metabolism in living organisms but not in metabolizing 3-HK. In mosquitoes though, HKT is the only enzyme that metabolizes 3-HK [11], which was first identified from *Aedes aegypti* in 1997 [12]. HKT is found to catalyze the transamination of 3-HK and L-kynurenine to xanthurenic acid and kynurenic acid, respectively. It also catalyzes the transamination of alanine with glyoxylate as an amino group acceptor, but less efficiently [10,13]. The detoxification function of the enzyme HKT over 3-HK makes HKT a potential insecticide target [14,15,16,17].

Interestingly, HKT is the only AGT homolog involved in detoxification of 3-HK, and it is only found in mosquitoes. In other insects and animals, such as mammals, kynureninase and L-kynurenine aminotransferases take the responsibility of metabolizing 3-HK. However, kynureninase is apparently absent in mosquitoes, just presenting L-kynurenine aminotransferases, which showed high activity for metabolizing kynurenine, but they cannot catalyze 3-HK [11]. Therefore, mosquitoes seem to deal with the detoxification of 3-HK using different genes, compared to other animals. In this particular case, the main biochemical function of one of the paralogous of AGT in mosquitoes, HKT, has changed. Previous studies hypothesized that the two copies of *AGT* genes may have originated and evolved in mosquitoes, due to the aquatic environment in which the larvae of mosquito dwells. The protein-rich food source of environment, resulting in a large portion of tryptophan being converted to 3-HK and, hence, in a greater demand for 3-HK detoxification [18]. However, little research has been done to understand the molecular biochemical evolution of HKT or AGT homologs in general. Accordingly, the evolutionary mechanism behind it remains unclear. 

Gene duplication is one of the principal processes in which new genes arise [19,20]. These new genetic products are the basis for the functional differentiation of homologous genes and are an important driving force in the evolution of genome and species [21]. Studying biochemical evolution of enzymes has emerged as a powerful tool for understanding their functions. Here, we provide the first comprehensive analysis of the molecular evolution of *HKT* and *AGT* genes. We found a gene-duplication event responsible for the two *AGT* paralogous genes in mosquitoes, which were retained in all species examined. Using the sequence comparison and computational analyses, we provide the first molecular evidence for divergence between *AGT* and *HKT* in mosquitoes after duplication of its common ancestral *AGT* homolog and suggest that positive selection after duplication lead to functional divergence. Reconstruction and molecular docking analysis of ancestral mosquito AGT (mosqAGT) revealed that 3-HK and kynurenine could serve as substrates for the ancestral mosqAGT. Our finding reveals the evolutionary mechanism by which HKT originated and provides an evolutionary example of neofunctionalization.

## 2. Results

### 2.1. HKT Phylogeny and Mosquito-Specific Duplication

To investigate the evolution of *HKT* gene, we search for homologous proteins by executing a BLASTP and tBLASTn search among different taxa using the HKT amino acid sequence of the mosquito *Aedes aegypti* (AaeHKT) as query. We found putative AGT homologous in other arthropods, as well as animals belonging to other phyla, and even in bacteria and Archaean species. We found that two AGT homologous proteins were present in the mosquito genomes among the 18 surveyed species (Appendix A), whereas other lineages, excluding mosquitoes, generally contain only one homolog of AGT. To characterize the degree of similarity of these proteins, and, therefore, predict their homology, we performed similarity statistics on the hits with the highest similarity for each taxon (Appendix A). From our data, we found that the highest similarity indices between AaeHKT and other organisms AGTs, such as other insects, mammals or bacteria, were between 42.4–55.6%. The best hit of the mosquito *A. albopictus* correspond to its HKT homolog and has the highest similarity with AaeHKT (95%). HKT and AGT from *A. aegypti* are highly divergent and share only 51.4% similarity, which is lower than the highest similarity hit between AaeHKT and AGT from beetle species, corresponding to 55.6%.

Due to the abnormally low similarity between HKT and AGT, 67 sequences were collected to reconstruct a phylogenetic tree to gain an initial idea of the evolutionary relationship between AGT homologs. To assess the robustness of the results, we reconstructed the gene phylogeny of AGTs using two different model-based approaches, Maximum Likelihood (ML) and Bayesian methods, and found it to have nearly identical topology (Figure 1, Appendix A). The topology shows some congruence with the phylogenetic species tree relationships, with AGT sequences falling in close relationship with sequences form members of the same clade (i.e., bacteria AGT sequences forming a clade). Mosquitoes HKT sequences together with AGT sequences from mosquitoes and other dipterans form a monophyletic group. These results suggest a vertical inheritance of both genes from a common mosquito ancestor, or in whole dipterans but lost in non-mosquito ones, instead of a horizontal gene-transfer event.

Next, to eliminate the potential biases caused by using distantly related species, we focused on analyzing AGT sequences from Insecta species (Figure 2, Appendix A). Both mosquitoes’ homologs appear as sister clades, but the one containing sequence labeled as AGT forming a clade containing also *Drosophila* sequences, whereas the one with sequences labeled as HKT forming a robust monophyletic group composed exclusively of mosquito lineages. These results suggest that mosquito sequences labeled as AGT might retain the ancestral AGT structure found in other insects, whereas the second clade containing the HKT sequences has diverged considerably. The sister group relationship between dipteran AGT and HKT clades reflects that they share a common ancestor, and may indicate a gene-duplication event in the early evolution of mosquitoes, or in the whole dipterans with non-mosquito lineages having lost one of the paralogs.

### 2.2. Micro-Synteny Analyses

To determine the physical location of *AGT* and *HKT* genes and its distribution on chromosomes, we conducted a syntenic analyses using the Genomicus database (Figure 3). The direction and arrangement of upstream and downstream genes of *HKT* in different species are similar, indicating that these genes have a significant syntenic relationship in mosquitoes. We also noticed that *HKT* and *AGT* in mosquitoes were located on different chromosomes, but there were no other homologous genes shared in the upstream and downstream nearby regions on the chromosome. The presence of genes in a shared syntenic location across species is a strong indicator that they are likely orthologous, which suggests that *HKT* and *mosqAGT* were both orthologous with other insect *AGT* and a single ancestral *mosqAGT* was present in the common ancestor of all mosquitoes.

Overall, our phylogenetic and synteny analyses together identify an independent duplication event that likely occurred during the history of mosquito evolution. Based on the chromosome location of *mosqAGT* and *HKT*, we speculate that a duplication of a genomic fragment occurred in the common ancestor of mosquitoes, resulting in the incorporation of that duplicated fragment into another chromosome. Along mosquito evolution, the newly formed *HKT* locus was retained, while other genes from the original duplicated strand of the chromosome were likely pseudogenized or lost.

### 2.3. Gene Structure Divergence between HKT and AGT in Mosquitoes

From the moment a new copy of a gene appears, divergence between the duplicates is the key to the preservation of new genes [21]. To reveal the divergence between *HKT* and *AGT* sequences of mosquito after duplication, we conducted a splice-site analysis (Figure 4). We found that both genes had different numbers of exons, with *AGT* having an average of three exons, while the number in *HKT* varies between three and six, highlighting divergences in the gene structure in the latter. Besides HKT from *Culicinae* and *Aedes*, both *AGT* and *HKT* contain an exon of 95 nt length. *Anophelinae* species *HKT* sequences contain an exon of 248 nt length, whereas *Culicinae HKT* contains an exon of 344 nt length. Interestingly, we have noticed that, although *Culicinae HKT* was lacking of exon of 248 nt length, the exon of 344 nt length was almost equivalent to the sum of two exons of 95 nt and 248 nt in the *Anopheles HKT,* suggesting that *Stegomyia HKT* was likely to have undergone exon-shuffling events. More narrowly, *Stegomyia HKT* lost one intron, leading to fusion of two adjacent exons. Splice-site analysis indicates that *HKT* and *AGT* experienced different intron gain/loss events. Changes in the structure of the coding region of the new copy are likely to affect the exon–intron structure after gene duplication, which makes functional divergence possible between *AGT* and *HKT* in mosquitoes.

### 2.4. Functional Divergence between AGT and HKT

To quantify the degree of functional divergence, we applied type I functional divergence analysis using DIVERGE 3.0 (Table 1). We used different branches for the pairwise comparison. As shown in the table, all detected functional divergence values are statistically significant. By comparing HKT and AGT in mosquitoes with HKT and Diptera AGT, HKT and Coleoptera AGT, HKT and Lepidoptera AGT, Dipteran AGT and Coleopteran AGT, the coefficients of functional divergence I (θ) were 0.288 ± 0.096, 0.292 ± 0.082, 0.385 ± 0.082, 0.056 ± 0.081, 0.533 ± 0.063, and 0.349 ± 0.063, respectively. These coefficients were all significantly > 0, indicating the functions between them have undergone significant functional divergence. Moreover, the coefficients between HKT and AGT from mosquitoes are smaller than those between HKT and other branches, suggesting a functional divergence of HKT in mosquitoes.

### 2.5. Detection of Positive Selection in HKT Genes

In order to detect whether *HKT* genes evolved under positive selection, we conducted positive selection analysis. Firstly, we executed a site model to preliminary screening for positive selection pressure. In the site model, by comparing the M0/M3 model, the result shows that the assumption of the M3 model was accepted, that is, it is assumed that ω values of all sites show a simple discrete distribution trend. The comparison of M1a and M2a models shows that there were only conservative sites and neutral sites, but no positive selection sites. In the pairwise comparison of M7 and M8 models, the alternative hypothesis was rejected, which suggests that the ω of all site belongs to matrix (0,1) and presents beta distribution. Taken together, site model analysis indicated that most sites in datasets represent purifying selection during evolution.

Additionally, the branch model was used to judge whether a specific branch was affected by positive selection and whether its evolutionary rate was different from background branches. According to the comparison results of M0 model and free ratio model, the alternative hypothesis was accepted, indicating that each branch was subjected to different selective pressures. Then, we select *HKT* and the *mosqAGT* branches as the foreground branches for two-ratio model analysis. The results show that the selection pressure of both genes is significantly different from that of the background branch. Moreover, we noticed that in both the free-ratio model and the two-ratio model, the ω of the *HKT* branch was greater than 1, while the ω for the *mosqAGT* branch was less than 1, reflecting that *HKT* was under positive selection pressure, and *AGT* in mosquitoes was subjected to convincing purifying selection during evolution.

Finally, given that the ω of *HKT* is greater than 1, we used the *HKT* branch as the foreground branch for the branch-site-model calculation. This part is to repeatedly verify the branches under the positive selection pressure, as calculated by the branch model, and find out the specific sites that were affected by positive selection. A total of 375 potential positive selection sites were detected, with 41 of them being considered as significant positive selection sites, with the posterior probability of these being >95%. The above results indicate that each branch have been under different selection pressure in the process of evolution, and the *HKT* branch is under strong, positive selection pressure (ω = 999). At the same time, a large number of positive selection sites have been detected in *HKT*. These sites are likely to cause divergence of *HKT* and *mosqAGT.* The refinement results are summarized in Table 2.

### 2.6. Positive Selection Sites Affect Substrate Binding

As the sequence identity and biochemical function of HKT and AGT in mosquitoes seems to have greatly diverged, this phenomenon will inevitably be reflected in the tertiary structure of the protein. Meanwhile, we also detected many sites affected by positive selection pressure, but the relationship and distribution of these sites in the tertiary structure were not clear. Consequently, we obtained the crystal tertiary structure of *A. aegypti* HKT from the PDB database (PDB ID 6MFB). Their crystal structure shows that the enzyme is a dimer and belongs to the fold type I class of the PLP-enzyme family. Then, we performed a structural similarity analysis by heuristic searches of the Protein Data Bank (PDB) with Dali and TM-align servers, taking 6MFB as the query. Both programs identified 6MFB as the top hit, followed by *A. gambiae* HKT (PDB ID: 2CH1), *A. aegypti* AGT (PDB ID: 2HUU), human AGT (PDB ID 3R9A), *Mus musculus* AGT (PDB ID: 3KGX), and *Nostoc* sp. AGT (PDB ID IVJO). Quantitative comparisons values indicated that 6MFB with the other four structures are likely to adopt the same global protein fold (Appendix A).

Next, we mapped positive selection into the crystal tertiary structure of *A. aegypti* HKT (Figure 5A). Among the 41 putative positive selection sites, 39.0% of them are located in the α-helix region, 43.9% in the loop regions, and 17% in the β-sheet. We found that most of these sites were distributed around the active sites; only a few sites coincide with active residues (Figure 5A, Appendix A). In particular, S43 and F45 are closely binding with N44 (Figure 5B), which is considered as a critical site to bind 3-HK and L-kynurenine [17,22]. Some sites with a strong signature of positive selection, W355, C319, Q344, and Y372, are immediately adjacent to or very close to R356 (Figure 5C)**,** which is responsible for forming a salt bridge with the carboxyl group of the ligand [17]. The distribution of positive selection sites in these regions means the modification of ligand binding conformation, which affects the substrates binding ability and structural stability.

### 2.7. Ancestral Sequence Reconstruction of AGT in Mosquitoes

With the aim of understanding the evolutionary history of *HKT*, we performed ancestral state reconstruction analysis to infer the sequence of *AGT* before duplication in mosquitoes. A total of 22 well-annotated *HKT/putative HKT* (*puHKT*) and *AGT* sequences were used to infer the ancestral sequence composition. Ancestral sequence at the node corresponding to the last common ancestor of mosquitoes was inferred and named AncMosqAGT (Figure 6). The inferred AncMosqAGT sequence shares 74.17% amino acid sequence identity with modern *C. quinquefasciatus* AGT.

Next, we focused on the active residues that were possibly involved in substrate binding. Sequence alignment suggests that some residues matched the active sites in modern proteins (Appendix A). Some of these sites are well-conserved in both HKT and AGT in mosquitoes, including P5, G6, P37, W105, W165, V189, D190, V192, S214, Q215, K216, Y267, L358, and R367. Most of them are involved in binding or maintaining the conformation of PLP binding domain, except R367 and the P35-G36-P37 triplet, which are considered key residues for ligand binding. Some sites are conserved only in mosqAGTs, including G54-H55-L56, G89, and S355. Among them, G54-H55-L56 was mutated into an S43-N44-F45 triplet in HKT, which may be involved in the recognition of 3-HK and L-kynurenine [22]. G89 was replaced by A78 in HKT and might be involved in alanine recognition, while S355 was replaced by Q344 in HKT and may have played an important role in substrate recognition [17,22]. We also compared the relationship of positive selection sites in three cases (the ancestral protein, HKT, and AGT in mosquitoes) and found that HKT had undergone large changes. These results suggest that the common ancestral sequence of AGT was more similar to modern AGT than to HKT.

### 2.8. Homology Modeling, Molecular Docking, and Dynamics Simulation of AncMosqAGT

To explore the activity of the ancestral protein, I-TASSER was used to model a three-dimensional structure. Six templates were found with >28% identify of the whole template chains with query sequence and >82% coverage of the threading alignment (Appendix A). Multiple sequence alignment of templates and query reveal conserved regions in the primary structure (Figure 7). Five homology models were generated and ordered by C-score. The top 1 model was used to conduct energy minimization for further analysis. The optimized model was evaluated by SAVES sever. ERRAT shows an Overall Quality Factor of 94.8276. 89.37% of the residues have averaged 3D-1D score ≥ 0.2. Evaluation of the model with PROCHECK shows that 97.4% of residues are located in the most-favored and allowed regions, while 0.6% of residues were found in the outlier regions (Appendix A). Superposition of the final model and templates visualizes the similar fold between them (Appendix A). The root-mean-square deviation (RMSD) of the Cα atoms between the templates and the final model are between 0.50 Å–1.07 Å. These above results indicate that the homology model is reliable.

To understand the ligand capacity and the ligand-binding mechanism of AncMosqAGT, which provide inspiration for shedding light on the key active sites of HKT, AncMosqAGT·PLP was docked with three different substrates: 3-HK, alanine, and L-kynurenine. Remarkably, the three substrates were stably bound to the ancestral protein. The docking solutions of 3-HK, alanine and L-kynurenine resulted in binding affinities of −5.5 kcal/mol, −5.2 kcal/mol, and −3.6 kcal/mol, respectively. Then, three poses were selected to analyze 3D molecular interactions (Figure 8). The docking results show that R367 forms a strong salt bridge with the carboxyl group of 3-HK, locking the ligand in the active site. The amino group of G36 forms a hydrogen bond with the aromatic amino group of the ligand. K216, S355, and G356 also have hydrogen-bonding interactions with 3-HK. W104 forms a hydrophobic interaction with the ligand (Figure 8A). These bonds help stabilize the conformation between the substrate and the protein. We also visualized the residues of 3-HK within the 4Å distance range (Figure 9). These residues are also involved in ligand recognition in modern HKT. Similarly, in the docking result of the ancestral protein and L-kynurenine, R367 also form a salt bridge with L-kynurenine. K216, G356, S165, W115, and L358 form hydrogen-bonding networks, hydrophobic interactions, and π-stacking with L-kynurenine (Figure 8B). Ala also forms a salt-bridged bond with R367 (Figure 8C). Therefore, we reasonably consider that ancestral protein was not only capable of transamination using alanine as an amino-group donor, but also possessed the binding properties of 3-HK and L-kynurenine.

In order to test the reliability of the docked poses, the protein-ligand complex results from docking were validated with a 100 ns molecular dynamics (MD) simulation using Gromacs software. The docked poses remain structurally stable after 100 ns in neutral solvated system, since the RMSD gradually stabilize with the increase in simulation time (Figure 10). The complex of 3-HK and AncMosqAGT·PLP is consistently achieved at the RMSD, ~0.7 nm after 75 ns. The RMSD of L-kynurenine generally maintains a stable trend after ~65 ns, while the RMSD fluctuates slightly at 0.75 nm. The RMSD value of alanine’s docking complex fluctuates during the simulation process, but the deviation gradually trends to narrow after 90 ns (the fluctuation range of RMSD value is less than 0.2 nm).

## 3. Discussion

In this study, we investigated the evolutionary process of HKT through computational-biology approaches, which focused on the phylogenic relationship, selection pressure, and the docking analysis. Our findings revealed insight into the origin of HKT, and strong positive selection enhanced the specific functions for HKT after a gene duplication, which sheds new light on the research of a new functional-enzyme biochemical evolution.

We have interrogated homologs of AGT in Protozoa and Metazoa; the existence of homologues in a wide range of bacteria to humans proved that AGT was ancient and conserved. This result is in line with our expectations because AGT belongs to PLP (vitamin B6)-dependent enzymes, which has appeared in the early stage of biological evolution, and functional specialization seems to have occurred in the common ancestor cells before the divergence of Eukaryotes, Archaebacteria, and Eubacteria 1500 million years ago [23]. In addition, we found out that 18 mosquitoes species contained two copies of AGT. Based on the clustering in the phylogenetic tree, these second copies were confirmed as puHKT. Thus, it appears plausible that a single duplication event occurred, leading to the present phylogenetic distribution and their homologous relationship.

Our conserved syntenic analysis found *HKT* and *mosqAGT* in a shared syntenic location across species, which was a strong indicator that they are likely orthologous. The syntenic analysis, together with the above phylogenetic tree, further imply that the most reasonable explanation of *HKT* emergence, that is, *HKT* is formed via a segmental duplication event in the common ancestor of mosquitoes. A species-specific segmental duplication in ancestral mosquitoes between chromosomes gave rise to a duplicate chromosome fragment. However, only 18 species that contain *puHKT* were found, and this number may be limited by the lack of genomic data on mosquito species. Differences in the copy number in mosquito species suggest that the *HKT* gene has suffered extensive gene loss after speciation.

Retention of duplicated genes is usually accompanied by structural changes in regulatory or coding regions, further leading to functional divergence. We found that the gene structure of *HKT* was significantly different from that of *AGT*. In addition, we observed a *Stegomyia*-specific exon-shuffling event. Exon shuffling is one of the molecular mechanisms by which new genes are formed [24]. Exon–intron divergence between duplicate genes is common. Structural divergence caused by intron loss/gain plays an important role in duplicating genes and can result in the generation of proteins with distinct domain organization and sequence features, leading to the changing of biochemical functions [25]. Therefore, AGT orthologs may adapt to different functions by altering the regulatory and/or coding regions. Furthermore, we measured functional changes of the *HKT* and *mosqAGT* after duplication. We found that changing functional constraints occurred after duplication, resulting in functional divergence between *HKT* and *mosqAGT*, but they were still the closest structural ortholog and functional ortholog (Table 2, Appendix A). The previous study had proved the divergence of the biochemical functions between HKT and AGT in mosquitoes. Here, our study provides bioinformatics evidence of functional divergence for the first time, which is consistent with biochemical evidence.

Given the functional differences between the duplicated *AGT* paralogous, we hypothesized that *HKT* had undergone adaptive evolution and that the corresponding amino acid substitutions would be correlated with functional diversification and that positive selection would be detected. From our selection pressure analysis, the site model rejected the hypothesis that sites with ω > 1 existed in the dataset. We speculated that this might be because most sites undergo purifying selection in the datasets, resulting in neutralization of positive selection. Nonetheless, *HKT* shows sufficient evidence for positive selection in the branch model and branch-site model, and *mosqAGT* shows evidence for purifying selection during evolution. This result ties well with previous studies. *A. culicifacies* and *A. stephensi AGTs* shows a negative selection pressure against protein change [26].

Bayes Empirical Bayes (BEB) implemented in the branch-sites test identified a number of positively selected sites in *HKT* (Table 2). We discovered the positive selection affected the substrate binding and the spatial conformation of the protein, although binding sites are not the direct targets of positive selection. Positive selection of residues positioned near active sites may be the general mechanism of functional diversification of the enzyme family. Moreover, mutations close to the active site are more likely to cause subtle conformational changes in the substrate-binding pocket than mutations far away [27,28]. Future research on insecticide targets can conduct mutational analysis of these sites to find powerful targets that inhibit enzyme activity.

The classic model to explain the evolutionary mechanisms involved after duplication and split into paralogous genes, championed by Susumu Ohno, proposed that these events create redundant loci able to perform new functions (neofunctionalization) or functions that were previously nonexistent [29]. Nevertheless, other more recent views, such as Bergthorsson and colleagues [30], argued that situations in which paralogs retain different parts of the ancestral function (subfunctionalization) cannot explain the origin of de novo adaptive phenotypes. They proposed a model involving innovation, amplification, and divergence (IAD), in which new functions were acquired prior to duplication, and subfunctionalization occurred among subsequent duplicated genes [30,31]. Whether the retention of *HKT* and *AGT* paralogous in mosquitoes is more consistent with IAD or Ohno model depends on the function of AGT in species with a single copy and substrate-binding capacity of the ancestral protein. Clearly, alanine is a native substrate of AGT. Thus, whether the ancestral protein has ability to catalyze 3-HK and L-kynurenine had become our major concern. Our molecular docking result showed that AncMosqAGT had broader substrates specificity, so 3-HK and L-kynurenine were able to bind with ancestral proteins, which may be verified by an experimental test in future. MD simulation demonstrated the docked ligand–protein complexes were stable and reliable in binding to the active sites.

R and S-N-F triplet residues were previously considered as the critical and direct target for possibly binding of 3-HK and L-kynurenine in modern HKT. By comparing critical residues of deduced AncMosqAGT and modern sequences, we discovered that only R residue was present in the ancestral protein. However, based on the molecular docking results, ancestral protein is able to bind 3-HK and L-kynurenine. Thus, R residue appears to take on greater responsibility for binding 3-HK and L-kynurenine.

Actually, the prominent role of conserved R residue in active sites has been repeatedly reflected in our study. The region near R356 residue is subjected to strong positive selection pressure. This changing of residues may affect the side-chain conformation and molecular orientation of Arg. Molecular docking shows that R367 residue in the ancestor could interact with the carboxyl groups of the three substrates to form salt bridges. Previous studies have also demonstrated that R residue formed salt bridges with the carboxyl groups of different ligands in modern protein, including 3-HK, L-kynurenine, glyoxylic, alanine, and 3-HK analogs. R residue is highly conserved in ω-aminotransferase, including modern mosqAGT. A density functional theory (DFT) calculations study of the half-transamination of L-alanine to the pyruvate reaction in (S)-selective *Chromobacterium violaceum* revealed the dual-substrate recognition mechanism of flexible R residue. The binding with alanine or hydrophobic amino (S)-1-phenylethylamine depended on whether the side chain of the R residue was positioned inside or outside the active site [32]. A conserved R residue in ω-aminotransferase from *Paracoccus denitrificans* led to substrate promiscuity as the flexibility [33]. These findings suggest that R residue appears to have a vital role for substrate recognition by HKT and AGT in mosquitoes. Although the current evidence points to the unusual properties of the R residue, it is difficult to explain such results without site-directed mutation experiments.

Overall, our results support the conclusion that the evolution of *HKT* followed the IAD evolutionary model. The changing of the ecological niche causes the parental gene to first have a minor side activity, namely the transaminase activity towards 3-HK and L-kynurenine. Next, the level of minor activity increases through parental gene-duplication event. To improve the fitness and coordination of the two genes in mosquitoes, the minor function of one copy is amplified due to positive selection pressure. As one copy improves, the selection is relaxed for the remaining copy, which results in a loss of minor activity. Therefore, the extant mosqAGT and HKT play coordinated roles in mosquitoes, and the requirement for 3-HK detoxification at the larval stage and the requirement for 3-HK to maintain ocular pigmentation at the pupal stage and early adult stage provide a plausible explanation for the differential evolution of HKT and mosqAGT [18]. This is an example of an enzyme of the glyoxylate metabolism pathway being recruited to the tryptophan metabolism pathway by gene duplication.

## 4. Materials and Methods

### 4.1. Gene Identification and Phylogenetic Analyses

To identify AGT homologs, Tblastn and Blastp were used in NCBI (https://www.ncbi.nlm.nih.gov/, accessed on 6 October 2021), with AaeHKT as the query sequence. Sequences that had a query identity of >30% and an E-value of <1 × 10^−10^ were considered homology proteins. Pfam [34] and CD-Search [35] were used to identify the conserved domain. We selected amino acid sequences that were representative enough of evolutionary relationships to perform further analysis. Multiple amino acid sequence-alignment analyses were performed using MAFFT ver.7.313 [36], with the L-INS-i multiple alignment method and BLOSUM80 scoring matrix. ModelFinder [37] was applied to identify the optimal amino acid substitution model based on the Akaike Information Criterion (AIC) criterion. Then, IQ-TREE [38] ver.1.6.8 was used to build the phylogenetic trees with a standard bootstrap algorithm (1000 replicates) with the best fit model. Moreover, MrBayes [39] was conducted to build Bayesian trees, and the dataset ran for two million generations with four chains, until they reached converge at 0.05. When the phylogenetic trees inferred by the two methods are mostly consistent, the phylogenetic relationship is considered to be reliable. Evolview [40] server was used for visualization of the tree.

### 4.2. Conserved Micro-Synteny Analysis

Conserved synteny analysis is a powerful tool for establishing gene orthologous between species and investigating the duplication event. The syntenic relationship of AaeHKT and its up- and downstream genes, all its orthologous and paralogous copies in all the other sequenced metazoan genomes were revealed by Genomicus version 51.01 [41] from EnsemblMetazoa 51 database (http://metazoa.ensembl.org/index.html, accessed on 23 October 2021).

### 4.3. Sequence Structure Analysis

To reveal the sequence conservation and differences of *AGT* and *HKT*, the Vectorbase release 55 (https://vectorbase.org/vectorbase/app, accessed on 24 November 2021) and NCBI database (https://www.ncbi.nlm.nih.gov/, accessed on 24 November 2021) were used to obtain Generic Feature Format (GFF) or Generic Feature Format 3 (GFF3) file. The exon–intron structures were summarized by GFF/GFF3 file.

### 4.4. Analyses of Type I Functional Divergence

Type I functional divergence is defined as functional divergence resulting from changes at the rate of evolution. DIVERGE version 3.0 software (http://xungulab.com/software.html, accessed on 23 December 2021) was employed to test type I FD between different clusters [42,43]. The coefficient of functional divergence (θ) is a value that measures the functional divergence degree between two clusters according to the likelihood ratio test. θ significantly greater than 0 indicates functional differentiation between the two clades. This analysis involved 63 protein sequences that are the same as the dataset used in phylogenic analysis within Insecta.

### 4.5. Selective Pressure Analyses

To detect the selection pressure during the evolutionary history of *HKT*, EasycodeML v1.4 [44] was used to analyze the well-alignment of sequences and the phylogenetic tree built by the ML method as input. The sequences were aligned by the codon alignment model in MAFFT. ModelFinder was applied to identify the best-fit nucleotide substitution model based on the Akaike Information Criterion (AIC) criterion, and IQ-Tree was employed to build a phylogenetic tree. Branch-specific model was used to detect lineage-specific positive selection. The site-specific model was conducted to detect particular sites which are to be positively selected. Branch-site model was performed to detect positive selection that affects only some sites on pre-specified lineages. Each paired model was employed to be compared by the Likelihood Ratio Test (LRT).

### 4.6. Ancestral Sequence Reconstruction of the Common Ancestor of Culicdae AGT

Ancestral sequence reconstruction is a new, emerging protein-engineering strategy, which can calculate ancient protein sequences based on extant ones. By constructing ancestral sequences and combing experimental characterizations to solve a multitude of paramount biological questions [45]. For obtaining a most likely historically correct ancestral sequence, protein sequences were collected based on two protocols: First, the selected sequences were all experimentally verified or have a high-quality genome assembly. Second, the selected sequences were representative enough of evolutionary relationships [46] For collected sequences, MAFFT and Modelfinder were adopted for multiple sequence alignment and to identify a best-fit amino acid substitution model based on the AIC criterion. Then, MrBayes was performed to obtain a robust phylogenic tree. The *Tribolium castaneum* AGT and *Anoplophora glabripennis* AGT were set as the outgroup. Finally, ancestral sequence reconstruction was conducted with PAMLX [47,48] using the alignment sequences and the Bayesian tree as inputs. The ancestral sequence was extracted from the Codeml output files.

### 4.7. Homology Modeling, Molecular Docking, and Molecular Dynamic Simulation

The tertiary structures of ancestral mosqAGT were predicted by I-TASSER servers [49]. LOMETS approach was used to search for templates of similar folds from the PDB library. LOMETS is a meta-server threading approach containing multiple threading programs, where each threading program can generate tens of thousands of template alignments [50]. The templates of the highest significance in the threading alignments were used by the program, the significance of which were measured by the Z-score, which was defined as the energy score in standard deviation units relative to the statistical mean of all alignments [51]. Six templates were used to build homology models (Appendix A). Five homology models were generated and ordered by C-score (confidence score for estimating the quality of predicted models by I-TASSER). The top 1 model structure was selected to optimize using Swiss-PDB Viewer program [52]. Model quality was evaluated by Verify 3D, Errat and Ramachandran plot in the SAVES sever (https://saves.mbi.ucla.edu/, accessed on 16 July 2022). The coordinates of the cofactor PLP and the AncMosqAGT·PLP complex was obtained by superposing AaeHKT crystal structure (PDB ID 6MFB) onto the newly built model. Structural homology was analyzed by Dali [53] and TM-align [54]. The structure of substrates was obtained from NCBI Pubchem Database (https://pubchem.ncbi.nlm.nih.gov, accessed on 16 February 2022). Ligand and receptor were prepared with AutoDock Tools 1.5.6. (http://mgltools.scripps.edu/, accessed on 16 February 2022). AutoDock vina [55] was used for molecular docking for the substrates. The protein structure was treated as rigid, while the ligands rotatable bonds were permitted to rotate. A grid box (20 × 22 × 24 Å) spacing of 1.0 Å covered the entire active pocket. PLIP (https://plip-tool.biotec.tu-dresden.de/plip-web/plip/index, accessed on 16 February 2022) and Pymol (https://pymol.org/2/, accessed on 16 February 2022) was used to visualize the docking results.

Gromacs-2021 with the CHARMM36 force field [56] was used for simulation studies of all docked complexes. Topology and parameter files for ligands were generated by CHARMM General Force Field (CGenFF) program [57]. TIP3P water molecules were included and with neutralization by Cl^−^ ions. Minimization of the solvated structure performed by the steepest descent method until maximum force reached <1000.0 kJ/mol/nm at 300 K with constant pressure. V-rescale thermostat and Berendsen barostat [58] were set at 300 K and pressure at 1 bar for initially equilibrating the systems with a time step of 2 fs (NVT equilibration and NPT equilibration). Production MD simulation was run for 100 ns after equilibration at 1 bar and 300 K. Trajectories were saved and XMGRACE was used to analyze the results.

## 5. Conclusions

In the present study, we found that *HKT* was derived from a gene-duplication event in the mosquito ancestor and supported the hypothesis of an IAD gene-duplication model. This reflects an expanded requirement for detoxification of 3-HK in ancestral mosquitoes. After duplication, the two genes have suffered distinct evolutionary pressures, leading to a different gene structure and function. *HKT* experiences strong positive selection pressure, and the affected positive selection sites significantly shape the functional properties and expression patterns of *HKT*. *HKT* and *AGT* in mosquitoes have functionally diverged, but they display common protein folding and a similar catalytic mechanism, which indicate that they have the ability to strictly discriminate between their respective substrates. Future studies should focus on discovering the key residues that determine the substrate specificity. Our study reveals the molecular biochemical evolution mechanism of *HKT* for the first time and provides an example of enzyme recruitment in the evolution of a new function.

## Figures and Tables

**Figure 1 molecules-27-04929-f001:**
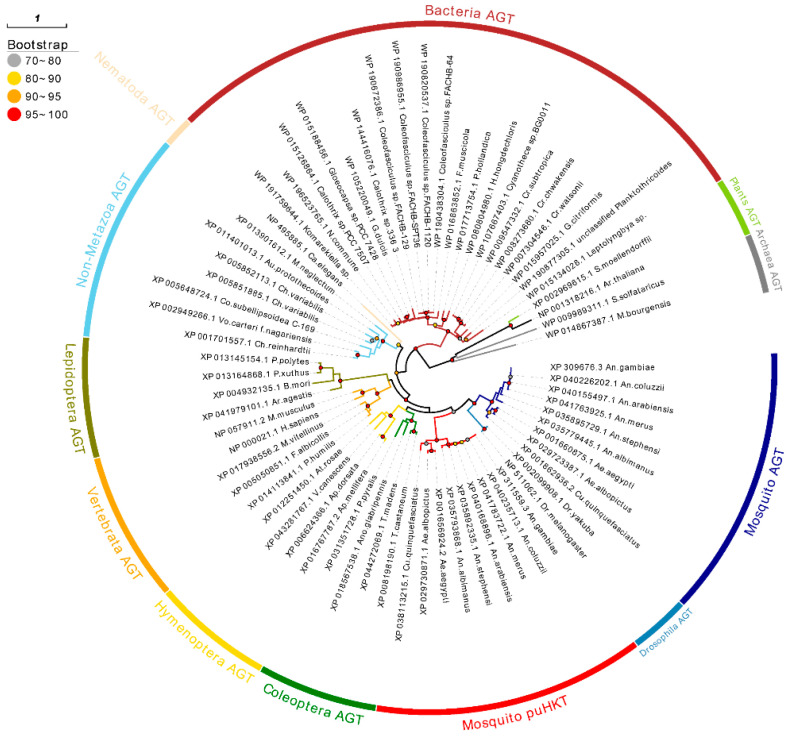
Evolutionary relationships among amino acid sequences of AGT. The evolutionary history was inferred by ML analysis described in Materials and Methods. Percentage ranges of bootstrap values of nodes were represented by circles in different colors. The tree was rooted by outgroup, which include two Archaeal species. Taxonomically related organisms are indicated by the same color code.

**Figure 2 molecules-27-04929-f002:**
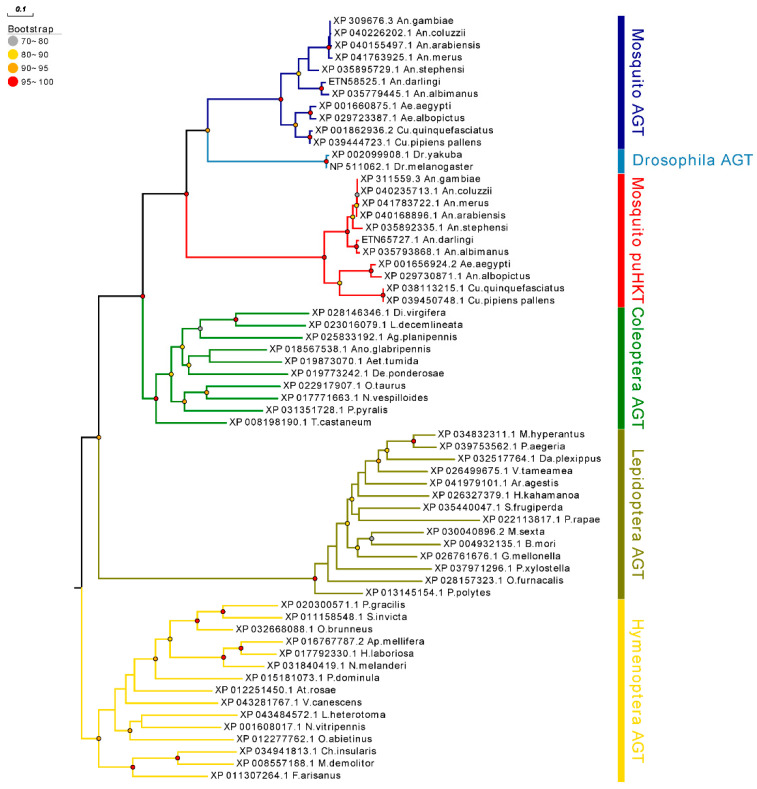
Detail of evolutionary relationships among amino acid sequences of AGT in Insecta. The evolutionary history was inferred by ML analysis described in Materials and Methods. Percentage ranges of bootstrap values of node were represented by circles in different colors. The tree was rooted by Hymenoptera AGT as outgroup, which are earlier in evolutionary history. Taxonomically related organisms are indicated by the same color code.

**Figure 3 molecules-27-04929-f003:**
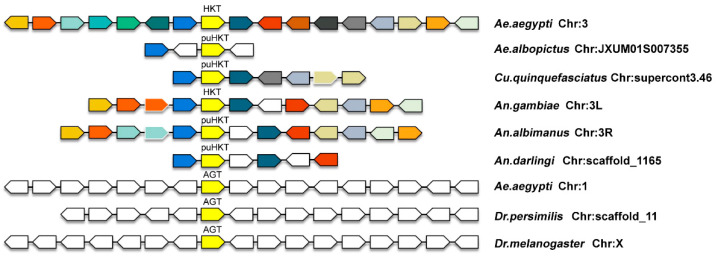
Syntenic analysis of mosquito *AGT* and *HKT*. Using AaeHKT as query, homologues of genomic genes located close to *HKT* were set within Diptera relatives. Genes were represented by an arrow label, the direction of the arrow indicates the direction of the gene on the chromosome, the same color markers represent same homologs, and the blank arrow represents a gene for which no homolog was found amongst different species. Location of the gene does not represent the actual proportion on the chromosome.

**Figure 4 molecules-27-04929-f004:**
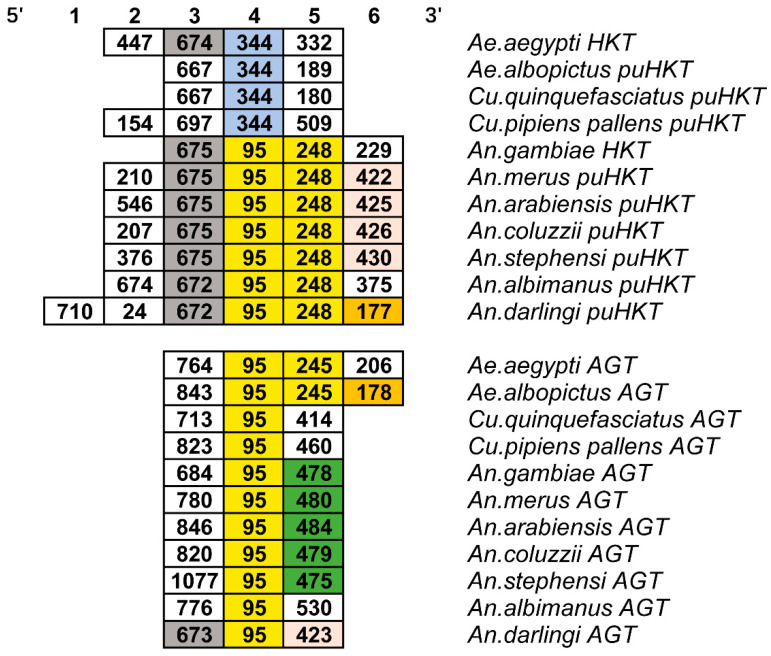
Exon structure of mosquito *AGT* and *HKT*. A box represents an exon, and numbers in boxes indicate exon lengths (nt). Same colors indicate similar exon lengths. The size of the box does not represent actual scale.

**Figure 5 molecules-27-04929-f005:**
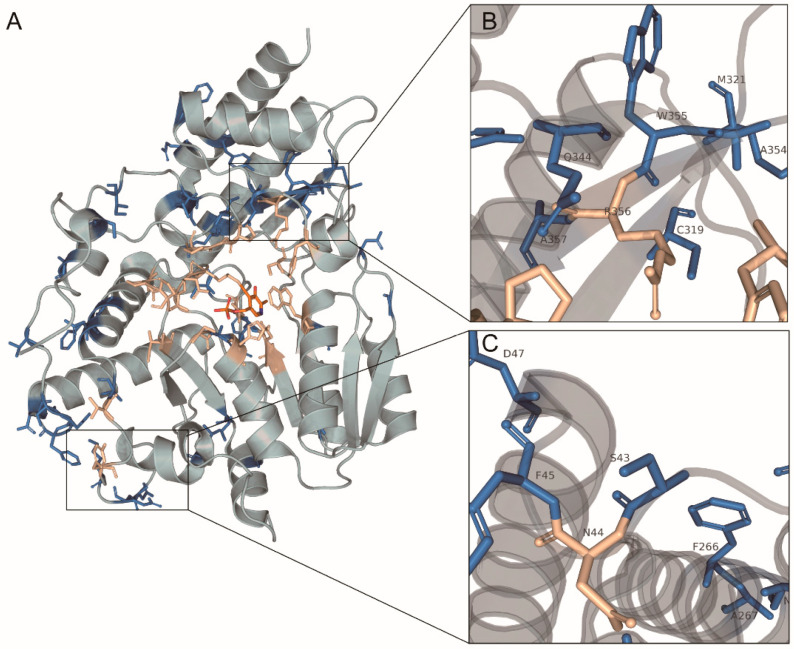
Distribution of positive selection sites on the three-dimensional structure of AaeHKT. The protein backbone structure is displayed in cartoon, and the key residues are shown in stick. Residues subject to positive selection are highlighted in blue, and residues contributed to the catalytic pocket are wheat-colored. PLP located in the hydrophobic cavity are colored orange. (**A**) Full view of positive selection sites on 3D structures. (**B**) Detailed view of R356 and nearby positive selection sites. (**C**) Detailed view of N44 and nearby positive selection sites.

**Figure 6 molecules-27-04929-f006:**
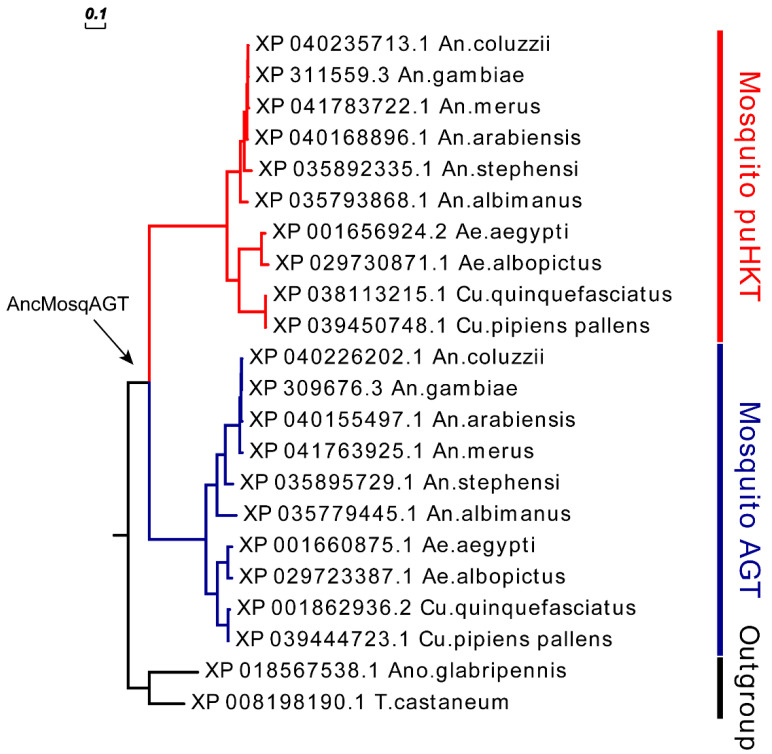
Phylogeny illustrating the last common ancestor of HKT and AGT in mosquitoes. The evolutionary history was inferred by Bayesian analysis. Black arrows point to the node of the last common ancestor of the inferred ancestral mosquito AGT.

**Figure 7 molecules-27-04929-f007:**
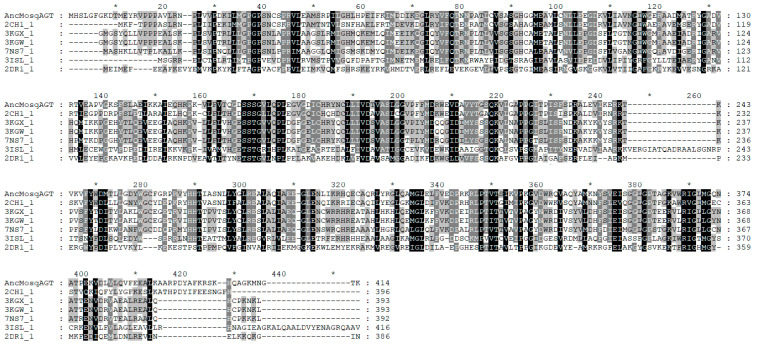
The sequence alignment of model with its templates from PDB database.

**Figure 8 molecules-27-04929-f008:**
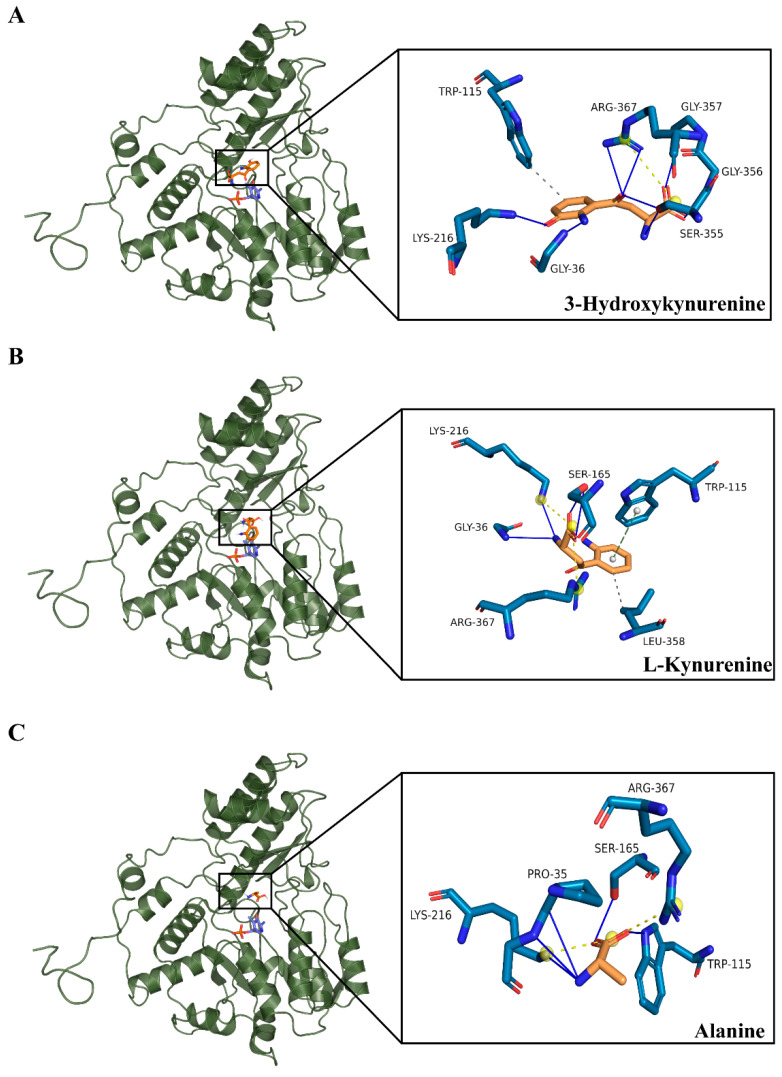
The 3D molecular interactions between AncMosqAGT·PLP and 3-HK (**A**), L-kynurenine (**B**), and Ala (**C**). Residues are shown as sticks and are different colored. Substrates are colored (orange for carbon atom, blue for nitrogen atom, and red for oxygen atom) as are key residues that interact with the substrate (darkblue for carbon atom, blue for nitrogen atom, and red for oxygen atom). Interactions are displayed using different forms (yellow dashed line for salt bridge, gray dashed line for hydrophobic interacts, blue solid line for hydrogen bonds, and green dashed lines for π-stacking.

**Figure 9 molecules-27-04929-f009:**
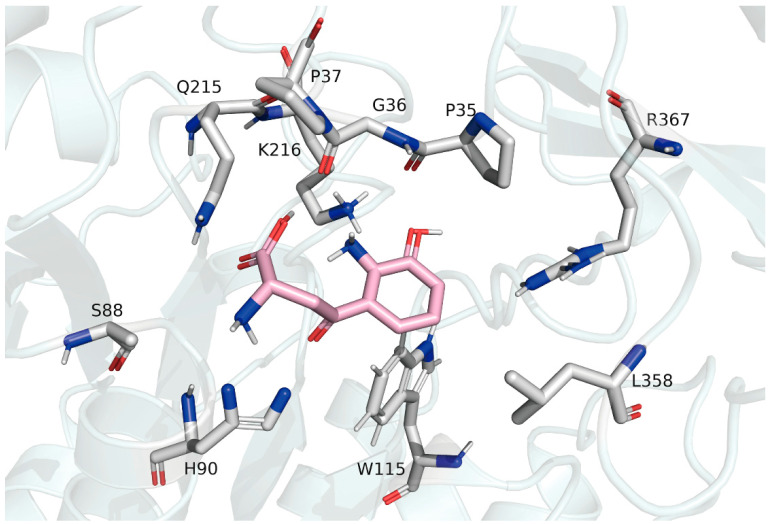
Residues in 4Å distance range from 3-HK, which is shown as stick and colored (pink for carbon atom, blue for nitrogen atom, red for oxygen atom, and gray for hydrogen atom). Residues in 4Å distance range from 3-HK are also colored (gray for carbon atom, blue for nitrogen atom, red for oxygen atom, and gray for hydrogen atom).

**Figure 10 molecules-27-04929-f010:**
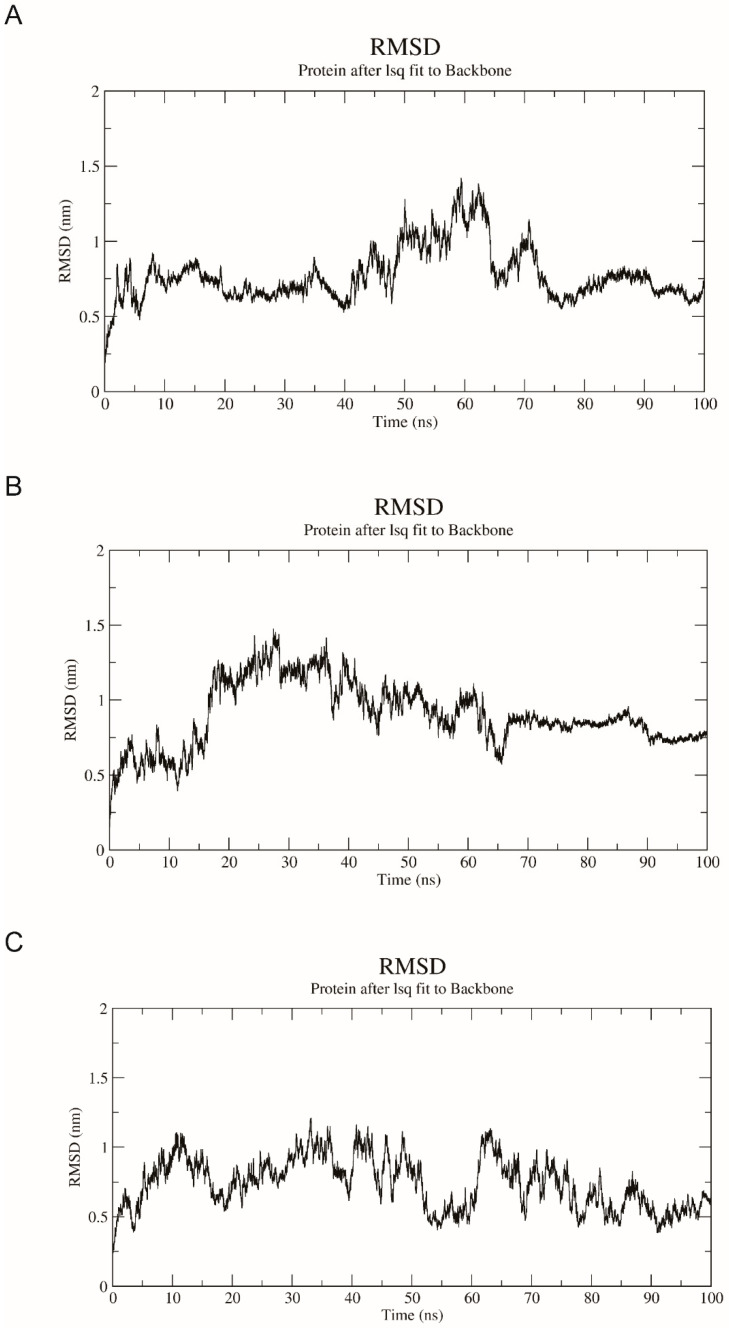
Backbone RMSD of the AncMosqAGT·PLP and 3-HK (**A**), L-kynurenine (**B**), and alanine (**C**) structures during 100 ns simulation. The ordinate is RMSD (nm), and the abscissa is time (ns).

**Table 1 molecules-27-04929-t001:** Type I functional divergence of AGTs in Insecta. The degree of functional divergence was quantified as the divergence coefficient θ, and the *p*-value was used to measure the significance of pairwise comparisons. DAGT, CAGT, and LAGT are short for Diptera AGT, Coleoptera AGT, and Lepidoptera AGT, respectively.

	Coefficient θ ± SE (*p*)
HKT/mosqAGT	0.288849 ± 0.096070 (*p* < 0.01)
HKT/DAGT	0.292432 ± 0.082008 (*p* < 0.01)
HKT/CAGT	0.385146 ± 0.082927 (*p* < 0.01)
HKT/LAGT	0.560887 ± 0.081839 (*p* < 0.01)
DAGT/LAGT	0.533485 ± 0.063676 (*p* < 0.01)
DAGT/CAGT	0.349754 ± 0.063676 (*p* < 0.01)
CAGT/LAGT	0.400981 ± 0.057970 (*p* < 0.01)

**Table 2 molecules-27-04929-t002:** Selection pressure analysis of *mosqAGT* and *HKT* by branch model, site model, and branch-site model. The ω represents for dN/dS. Lnl: log likelihood; LRT: likelihood-ratio test; 2Δlnl: twice the log-likelihood difference of the models compared. Sites labeled with * represent Bayesian posterior probability of >90% and labeled with ** represent Bayesian posterior probability of 99%.

Site Model							
Foreground Branch	Models	−lnl	2Δlnl	LRT Pairs	LRT *p*-Value	Estimates of Parameters	Positive Sites (BEB)
	M0	45,733.91409	2469.602752	M0/M3	<0.01	ω = 0.07927	Not allowed
	M3	44,499.11272				ω0 = 0.01224, ω1 = 0.07281, ω2 = 0.19935, p0 = 0.29651, p1 = 0.45743, p2 = 0.24606	Not allowed
	M1a	45,672.02204	0	M1a/M2a	>0.05	ω0 = 0.08631, ω1 = 1.0000, p0 = 0.96868, p1 = 0.0.03132	Not allowed
	M2a	45,672.02204				ω0 = 0.8631, ω1 = 1.0000, ω2 = 1.0000, p0 = 0.96868, p1 = 0.01322, p2 = 0.01810	None
	M7	44,510.89475	0.002114	M7/M8	>0.05	P = 0.77106, q = 6.60910	Not allowed
	M8	44,510.8958				p0 = 0.99999, p = 0.77108, q = 6.60934, p1 = 0.00001, ω = 1.0000	None
**Branch Model**							
**Foreground Branch**	**Models**	**−lnl**	**2Δlnl**	**LRT Pairs**	**LRT *p*-Value**	**Estimates of Parameters**	**Positive Sites (BEB)**
	M0	45,733.91409	658.98236	M0/Free ratio model	<0.01	ω = 0.07927	Not allowed
	Free Ratio Model	45,404.42291				ωmosqAGT = 0.3319, ωHKT = 999.0000, (variable ω)	Not allowed
HKT	Two Ratio Model	45,723.40152	8.104116	M0/Two ratio Model	<0.01	ω0 = 0.08006, ω1 = 999.00000	Not allowed
mosqAGT	Two Ratio Model	45,728.8162	10.195778	M0/Two ratio Model	<0.01	ω0 = 0.07926, ω1 = 0.00163	Not allowed
**Branch-Site Model**							
**Foreground Branch**	**Models**	**−lnl**	**2Δlnl**	**LRT Pairs**	**LRT *p*-Value**	**Estimates of Parameters**	**Positive Sites (BEB)**
HKT	M0	45,659.52254	8.718492	M0/MA	<0.01	ω0 = 0.08764, ω1 = 1.00000, ω2 = 1.00000	Not allowed
	MA	45,655.1633				ω0 = 0.08847, ω1 = 1.00, ω2 = 999.00	9S *, 16I *, 21M *, 31K *, 35A *, 40T *, 43S *, 44N *, 45F *, 47D *, 78A *, 105A *, 124G *, 127D *, 146C **, 180A *, 185C *, 190Y *, 218I *, 221K *, 240L *, 251D *, 252E *, 254K *, 260V *, 263N *, 266F *, 267A *, 286R *, 319C **, 321M *, 339F *, 344Q **, 354A *, 355W **, 357A *, 359I *, 363S *, 364S *, 367Q *, 372Y **

## Data Availability

Ligand docked structures and the structure of AncMosqAGT are available from the authors.

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
