# Peer review of "Biochemical Evolution of a Potent Target of Mosquito Larvicide, 3-Hydroxykynurenine Transaminase"

_molecules, 2022, doi:10.3390/molecules27154929_

Round 1

Reviewer 1 Report

The current work entitled “Biochemical evolution of a potent target of mosquito larvicide, 2 3-hydroxykynurenine transaminase” by Chen et al revealed the molecular biochemical evolution mechanism of HKT for the first time. The work is well written and presented and should be of interest to the journal readers. Few comments should be addressed before further processing for the manuscript.

The introduction should be more concise.

Authors should address the validation issue of the Homology modelling and Molecular docking studies.

The language of the manuscript needs a revision.

I suggest moving Figure S5 to the main body of the manuscript.

Reviewer 2 Report

Chen et al investigated evolutionary differences in 3-hydroxykynurenine transaminase (HKT) in a computational analysis, which could potentially be useful in mosquito larvicide studies. The manuscript's introduction is scientific, but the results provide minimal information. The authors provided computational information without empirical evidence, which is very useful for designing studies but does not draw conclusions. Accordingly, I do not support the publication of this manuscript.

1. What is the main question addressed by the research? The authors rely solely on computational analysis for their findings. The author's claim is unscientific because there is no experimental evidence that the author's claim that HKT is a potential drug target for mosquito larvicide. 2. Do you consider the topic original or relevant in the field? Does it address a specific gap in the field? Although the research topic has some originality, since there is no experimental evidence, it is judged that it is closer to a hypothesis through computer analysis rather than a research result. 3. What does it add to the subject area compared with other published material? It is interesting in terms of computational analysis, but it is difficult to compare it with other studies due to the lack of experimental evidence. 4. What specific improvements should the authors consider regarding the methodology? What further controls should be considered? It is the result of amino acid analysis using a combination of previously known programs rather than a new methodological attempt. 5. Are the conclusions consistent with the evidence and arguments presented and do they address the main question posed? Because no experimental results are presented, the authors' claims are more like hypotheses based on computer analysis. 6. Are the references appropriate? It is considered appropriate. 7. Please include any additional comments on the tables and figures. Since they are simple amino acid analysis results, they do not provide much information to the reader.

Reviewer 3 Report

Chen et al. entitled manuscript, "Biochemical evolution of a potent target of mosquito larvicide, 3-hydroxykynurenine transaminase," presented 3-hydroxykynurenine transaminase (3HKT) as an antimalarial target where they performed phylogenetic studies and constructed a homology modeling.

The key highlight of the paper is proposing 3HKT as a druggable target. Furthermore, justifying the genetic link with other genes and the three-dimensional modeling of 3HKT protein.

Some missing information and points need to address.

1.         Typographical errors, Page 1 line 12, tryptophan metabolic pathway. <space><space>The HKT enzy-; Please simply the complex sentences into simpler ones.

2.         The authors performed molecular docking and Homology modeling, but no explanation or figures were provided in the result and discussion of the paper.

3.         Significant information regarding homology modeling is missing

a)         Please add more information to the experimental section "How the template was searched for Homology modeling (what  identity and similarity was found with the template; how much sequence of the template was covered with the query sequence)"

b)         Alignment of the query sequence with a template sequence. Please add a Figure.

c)         In homology modeling, it is expected that some regions are not covered by template (also called Indels); how do authors address such issues during their homology modeling experiments; provide with an explanation?

d)         The authors mentioned in the abstract that "the homology model along with 3-hydroxykynurenine, kynurenine and alanine were used in docking experiments to predict the binding capacity and ligand-binding mode of the new substrates related to toxic metabolites detoxification" Which section was related in the main manuscript with these lines. Please add enough information to the main manuscript, such as 2D or 3D docking figures, ligand binding energies, and other parameters (10.1016/j.bioorg.2018.12.003)

e)         Parameters used for homology modeling and their qualitative analysis (10.1016/j.ejmech.2019.04.064).

a.         Ramachandran plot

b.         Superpose of homology model with the template (a 3D figures)

must be incorporated

The manuscript has sufficient elements that makes it fit in the current journal if the author agrees to revise the manuscript based on raised points.

Round 2

Reviewer 2 Report

I appreciate the author's sincere reply. I still find the subject of the manuscript interesting and valuable. Most of the manuscript results are analyzed using an existing computer program. I think this study will be a useful starting point for conducting future research, but there is no experimental evidence for this. Accordingly, I cannot support the publication of the revised manuscript.

1. The text related to the following subtitles is missing from the manuscript.

- 2.5. Detection of positive selection in HKT genes
- 2.6. Positive selection sites affect substrate binding
- 2.7. Ancestral sequence reconstruction of AGT in mosquitoes
- 2.8 Homology modelling, molecular docking and dynamics simulation of AncMosqAGT

2. Supporting data should consist of a separate file from the main manuscript.

Author Response

Dear Sir/Madam,

Thank you again for pointing out the weakness of our paper. I agree with you that if we would have provided experimental evidence, this paper will be a more solid one. However, considering the computational biology becomes a more powerful tool, which could answer more sophisticated questions alone. The computational results are accepted by many journals. We double-checked whole procedures we used in the paper, all were properly used, and the analyses support our conclusion, which you also thought interesting and valuable. In addition, the leading author whose background is bioinformatics graduated this year, I assigned a new student to do experimental work, based on the student plan, it will take one year to get the work done. The experimental work could be an independent story, which I believe will cite and support the conclusion of this paper. Taking together, I hope Molecules could consider this paper. Thank you very much!

Others:

  1. The text related to the following subtitles is missing from the manuscript.

- 2.5. Detection of positive selection in HKT genes

- 2.6. Positive selection sites affect substrate binding

- 2.7. Ancestral sequence reconstruction of AGT in mosquitoes

- 2.8 Homology modelling, molecular docking and dynamics simulation of AncMosqAGT

Regarding this concern, we noticed the texts were hidden after the titles. When the triangle link was clicked, the text will be appeared.

  1. Supporting data should consist of a separate file from the main manuscript.

We separated the file. Note that the refs indexed in the supporting data were changed accordingly.

Thank very much again for your valuable comments.

Best regard,

Qian Han

School of Life Sciences

Hainan University

Round 3

Reviewer 2 Report

It is judged that the manuscript is being gradually improved. Although the manuscript still lacks experimental evidence, I support the publication of the revised manuscript based on the opinions of other reviewers.